# Time of Postharvest Ethylene Treatments Affects Phenols, Anthocyanins, and Volatile Compounds of Cesanese Red Wine Grape

**DOI:** 10.3390/foods10020322

**Published:** 2021-02-03

**Authors:** Diana De Santis, Andrea Bellincontro, Roberto Forniti, Rinaldo Botondi

**Affiliations:** Department for Innovation in Biological, Agro-Food and Forest Systems–DIBAF University of Tuscia, 01100 Viterbo, Italy; desdiana@unitus.it (D.D.S.); forniti@unitus.it (R.F.); rbotondi@unitus.it (R.B.)

**Keywords:** Cesanese wine grape, ethylene, phenols, anthocyanins, volatile compounds, PCA

## Abstract

Red Cesanese wine grapes, picked at around 22–23 °Brix, were treated with gas ethylene (500 mg L^−1^) for 15, 24, and 36 h, or air at 20 °C and 95–100% relative humidity (R.H.), then analysed for titratable acidity, sugar content, pH, total phenols, total and specific anthocyanins, and volatile compounds. Ethylene treatments increased the polyphenol content from 412 to 505 and 488 mg L^−1^ (about +23 and +19%) for 15 and 24 h samples, respectively. Anthocyanins were increased by ethylene, mainly for 15 h treatment (about +17%). The 36 h ethylene treatment induced a loss anthocyanins (−14%), while phenols practically returned to the initial content. A high content of ethanol, acetic acid, and ethyl acetate were detected in 36 h ethylene-treated grapes, together with higher isoamyl acetate content, compared to air and other ethylene treatments. C6 compounds, markers of lipids peroxidation, were slightly higher in 36 h ethylene-treated samples than in control. Shorter ethylene treatments did not significantly modify the aroma profile compared to air treatment.

## 1. Introduction

Ethylene is a small, readily diffusible phytohormone which plays an important role in integrating developmental signals and responses to biotic and abiotic external stimuli. Ethylene is a critical component of such diverse developmental processes as seed germination, fruit ripening, abscission, and senescence. It is also widely viewed as a stress hormone. Adverse biotic or abiotic cues usually stimulate ethylene synthesis. At gene expression level, ethylene has been shown to induce transcription of a wide range of genes involved in wound signalling [1], pathogen defence [2], and fruit ripening [3].

Grape is considered a typical non-climacteric fruit, which means that ethylene production does not increase during ripening and not all the ripening features (colour, softening, sugar increase, acidity decrease) respond to exogenous ethylene treatment. In grape, ethephon and ethylene field treatments are known to increase colour and decrease acidity [4,5,6,7,8]. The potential role of ethylene in the induction of anthocyanin synthesis in wine grapes has been elucidated [9]. Chervin et al. [10], using 1-methylciclopropene (1-MCP) on a Cabernet Sauvignon grapevine at various times following full bloom, noted an inhibition of berry enlargement, an increase in acidity, and a transient inhibition of anthocyanin accumulation in berry skin, hypothesising a role of ethylene in the berry, and observing that ethylene application at veraison led to a berry diameter increase due to sap intake and cell wall modifications, enabling cell elongation. In 2016, Li et al. [11] well described the action of 1-MCP as ethylene antagonist even in non-climacteric fruits, including grape, so demonstrating the role of exogenous ethylene as promoter of several mechanisms related to the ripening processes. A paper recently published regarding Moldova grape berries studied the berry ripening under the effect of ethylene, which was in turn promoted by melatonin stimulation [12]. In a pioneering paper for postharvest applications, Bellincontro et al. [13], using postharvest ethylene (500 mg L^−1^ for 15 h at 20 °C) and 1-MCP (1 mg L^−1^ for 15 h at 20 °C) treatments on Aleatico wine grapes, have shown a significant increase in phenol content in berries and a stable anthocyanin content after six days from the treatment, together with a significant change in aromatic organic compounds. The same positive effect on polyphenols has been found by Becatti et al. [14] in red wine grape Sangiovese, treated for 36 h with ethylene (1000 mg L^−1^) before vinification. Always in Sangiovese variety, following the same protocol of treatment previously described, the effect of postharvest ethylene application has been demonstrated [15]. This result was associated to the mediating effect of abscisic acid (ABA) even related to the carotenoid accumulation, already described in grape berries by Sun et al. [16]. The same authors [16] have underlined the ethylene action in increasing secondary metabolites extractability in wine grape juice/must, as well as already demonstrated by Botondi et al. [17]. In cv. Verdejo, González et al. [18] found an improvement of the wine aroma by treating grape with ethephon at 1500 mg L^−1^ sprayed on bunches at veraison stage.

Cesanese is a red wine grape variety typically cultivated in the southern area of Latium Region in Italy, and it is known for low polyphenol content, the difficulty of their extraction at commercial maturity, and for passing poor varietal aromatic profile in related wines. Based on these assumptions, in the present paper, we hypothesised that ethylene postharvest treatment on Cesanese grapes for different exposure time could increase the polyphenol content and its extraction, and also improve the aroma in grape.

## 2. Materials and Methods

### 2.1. Test Material

Cesanese grapes (*Vitis vinifera* L. cv Cesanese) were collected from a vineyard located in the southern area of Latium Region (41°43′35″ N, 13°08′23″ E, altitude 573 m), grown to cordon-trained and spur-pruned; row and vine spacing were 2.6 m and 1.2 m, respectively, with north-south row orientation. Sound and uniformly coloured bunches with berries having refractometer index (RI) of 23.2 (±0.9) °Brix and 4.6 (±0.2) g L^−1^ of titratable acidity (TA), were harvested and carefully placed into 24 perforated plastic boxes (6 per treatment, about 6.5 kg per box). Grapes were picked early in the morning and treatments were started late in the afternoon.

### 2.2. Treatments

Each lot of 6 boxes (about 39 kg of grape bunches) was placed in 300 L airtight, stainless steel chambers located inside a temperature-controlled room for the following gas treatments, which are also reported in Table 1:500 mg L^−1^ gas ethylene, chamber sealed for 15 h, then opened, air ventilated and sealed again for 21 h to reach the same treatment time as treatment 3;500 mg L^−1^ ethylene, chamber sealed for 24 h, then opened, air ventilated and sealed again for 12 h to reach the same treatment time as treatment 3;500 mg L^−1^ ethylene, chamber sealed for 36 h;Air treated grapes, chamber sealed for 36 h.

Gaseous ethylene, at the purity condition of 99.9% N3.0 (CP grade), was picked from the original cylinder by the pressure reducer and flowed into the treatment chambers (Appendix A).

The reason for using 500 mg L^−1^ ethylene and also the use of 15 h treatment (n. 1) was due to the need to compare the basic treatments previously used on the Aleatico variety [13]. All treatments were performed at 20 °C (±1 °C); high relative humidity (95–100%) was maintained, as described by Bellincontro et al. [13]. Ethylene was supplied by a gas tank of 500 mg L^−1^ ethylene in air (Rivoira, Terni, Italy). CO_2_ accumulation was avoided by absorbing it on calcium hydroxide accurately placed inside the chambers. Since the high ratio free volume-grape mass, the accumulation of CO_2_ and the consumption of O_2_ was low in all the samples. A gas measurement was performed by using an Oxycarb device (Isolcell, Laives (BZ), Italy) based on electrochemical, or infrared detections for O_2_ or CO_2_, respectively; gas concentration was expressed as %. After 36 h, at the end of treatment, about of 14% of O_2_ was detected for all the chambers, while the amount of the residual CO_2_ ranged between 1.5 and 1.8%.

### 2.3. Quality Analyses

The juice obtained by squeezing 30 berries coming from 5 bunches (in triplicate) was used for the determination of soluble solids content (SSC), which was measured by a table refractometer model RL-2 (Abbè, Officine Galileo, Florence, Italy) calibrated at 20 °C, and reported as refractometric index (RI). Titratable acidity (TA) was measured through the titration of 5 g of the same juice, to pH 8.1 with 0.1 N NaOH, using a pH-meter model pH 300 (Hanna Instruments S.r.l, Ronchi di Villafranca Padovana (PD), Italy). Total polyphenols were determined using the Folin-Ciocalteau method and UV-Vis spectrophotometry was used to determine both total polyphenols and anthocyanins as described in Di Stefano et al. [19], where even extraction procedures were reported. All chemical analyses were performed in and the results expressed as averaged values ± the standard deviation (SD).

#### 2.3.1. HPLC Detections

A total of 50 g of skin berries, in triplicate, were collected and added, separately, with 100 mL of distilled water or 60% (*v*/*v*) of ethanol (96° RPE, Carlo Erba, Milan, Italy), homogenised by using an ultrasound bath for 5 min; then the extracts were centrifuged at 4 °C, 21,074× *g* for 15 min. The supernatants were collected and analysed by HPLC. The aqueous and ethanolic extracts were characterised by liquid phase chromatography reverse using a Dionex chromatograph (Dionex Corporation, Sunnyvale, CA, USA), equipped with P680 quaternary pump, manual injector (Rheodyne) with 20 mL loop, TCC-100 thermostated oven, PDA 100 detector (Photodiode Array Detector) and controlled from the Chromeleon software (version 6.50). The separation was carried out with a Dionex Acclaim^®^ 120 C18 column, 5 μm, 4.6 × 250 mm thermostated at 30 °C. The mobile phase consisted of a ternary gradient: solvent A = 50 mM ammonium dihydrogen phosphate adjusted to pH 2.6 with acid phosphoric; solvent B = 20% solvent A and 80% acetonitrile; solvent C = 0.2 M orthophosphoric acid adjusted to pH 1.5 with NaOH. The phenolic compounds were identified based on their elution order, the retention times of pure compounds and the characteristics of their UV-Vis spectra at the wavelength of 520 nm for anthocyanins. A semiquantitative evaluation of anthocyanins-3-monoglucosides was performed using chemical standards of cyanidin-, malvidin-, and peonidin- (Extrasynthese, Genay, France), delphinidin- and petunidin- (Polyphenols Laboratoires, Sandnes, Norway). The concentration of acylates and dimer compounds was computed using a response factor of malvidin-3-glucosyde.

#### 2.3.2. Volatiles

Volatile organic compounds (VOCs) were analysed by gas chromatography using solid phase microextraction, as described by Costantini et al. [20], slightly modified. A total of 5 mL of grape berry juice with 5 mL saturated CaCl_2_ added (1:1 *w*/*v*) was homogenised with 200 µL of standard solution of 1-penten-3-one (5 g L^−1^ in milli-q water) in a 25 mL glass mini-flask (Supelco, Sigma-Aldrich Co., St Louis, MO, USA), equipped with a small magnetic stirring bar, capped with a polytetrafluoroethylene (PTFE)-faced silicone septum, and placed in a thermostated bath under continuous stirring, for 15 min at 20 ± 2 °C. Then, the fibre, 100 µm polydimethylsiloxane (PDMS) (Supelco Inc., Bellafonte, PA, USA), was inserted into the flask headspace for 30 min. The fibre was conditioned in the gas chromatography (GC) injection port at 250 °C for 2 h prior to use. After the selected extraction time, the solid phase microextraction (SPME) fibre was transferred to the injection port and thermally desorbed at 230 °C for 7 min. The splitless injector was mounted on a Trace GC, ThermoFinnigan UltraGC (ThermoFinnigan Inc., San Jose, CA, USA) equipped with a fused silica capillary column impregnated with a polar phase of carbowax 20 m (Alltech Associates Inc, Deerfield, IL, USA), 60 m long × 0.25 mm ID. and 0.25 µm film thickness. Helium was used as carrier gas (27 cm s^−1^). The temperature was maintained at 40 °C for 7 min and then programmed to reach 230 °C at a rate of 3 °C per min, with a final isotherm of 30 min. A high sensitivity flame ionisation detector (FID) at 260 °C was used. The signal was recorded and integrated by a Mega Series integrator. Compound identification was achieved using a Shimadzu 17A GC-MS and a Shimadzu QP 5050A MS and matching against the NIST 107 and NIST 21 libraries, and by matching GC retention times against standards. The authentic standards for VOC identification and quantification were provided by Sigma-Aldrich (Merck Life Science S.r.l., Milan, Italy)

### 2.4. Statistical Analysis

ANOVA was performed for each quality parameter used to evaluate the effect of treatment and sampling time. The least significant difference (LSD) was calculated for the appropriate level of interaction. Inferential statistic and multivariate calculations were carried out by using Matlab R2015a (The MathWorks Inc., USA) software and related PLS Toolbox v. 8.0 (Eigenvector Research, Inc., Manson, WA, USA) application.

## 3. Results and Discussion

RI decreased, significantly, in ethylene-treated but also in air-treated grapes compared to the initial sample (Table 2), while titratable acidity and pH were not affected by ethylene treatments (Table 2). The observed sugar decrease must be due to a postharvest stress with an increase of sugar catabolism, mainly respiration. Total polyphenols increased significantly (about 23%) in grapes ethylene-treated for 15 and 24 h (from 412 mg L^−1^ up to 505 and 488 mg L^−1^, respectively) (Table 2). Ethylene treatment for 36 h maintained the initial concentration of polyphenols as well as air-treated grapes. Total anthocyanins significantly increased from 160 up to 187 mg L^−1^ in 15 h-ethylene-treated grapes, while they slightly decreased in 24, and significantly decreased in 36 h-ethylene-treated ones (Table 2). Control grapes significantly lost anthocyanin compounds (107 mg L^−1^). Checking at the anthocyanin/polyphenol ratio, which is important in terms of wine colour tonality and stability, it is possible to observe how ethylene-treated samples presented values similar to the initial ones, while in control grapes the values decreased (Table 2).

It appears that the stimulation of polyphenols synthesis by ethylene is transient, occurring in the first 24 h and disappearing later on. 15 h ethylene treatment appears to be the best treatment, inducing a significant increase in total polyphenols and total anthocyanins, while 24 h treatment stimulates the synthesis of polyphenols but not of anthocyanins and, finally, 36 h treatment induces the loss of both compounds. Regarding the specific anthocyanins, delphinidin-3-monoglucoside increased significantly in 15 and 24 h-ethylene-treated ones, where the content rose from 10 mg L^−1^ up to 15.8 and 15 mg L^−1^, respectively (Figure 1). Even petunidin-3-mononucleoside rose significantly from 11 mg L^−1^ up to 16.5 in 15 and 24 h ethylene-treated samples (Figure 1). Cyanidin-3-monoglucoside had the lowest content (4 mg L^−1^ initial concentration) and increased slightly only in 15 h-treated samples whereas peonidin-3-monoglucoside increased significantly from 10.4 up to 13.8 mg L^−1^ in 24 h-treated grapes (Figure 1). Acylated anthocyanins and dimers did not change significantly among the samples.

The increase of malvidin-3-monoglucoside was not as great in percentage (8% vs. about 30–45%) as the one of delphinidin and petunidin, rising from 37.5 up to 46 mg L^–1^ after 24 h of ethylene treatment (Figure 2). All the anthocyanins decreased in 36 h ethylene-treated samples to the same values of control and initial samples.

In a previous paper, Bellincontro et al. [13] showed that postharvest ethylene treatment on grape var. Aleatico stimulates the synthesis of polyphenols and partially of anthocyanins, confirming what observed by El-Kereamy et al. [9] with CEPA (2-chloro-ethylenphosphonic acid) treatment on the vine at veraison of Cabernet Sauvignon grapes. In Cesanese, short time postharvest ethylene treatment confirms the role of ethylene in the polyphenol and anthocyanin synthesis.

Regarding volatile compounds, our attention was addressed to the compounds which were more discriminatory among the samples, and the percentages increase or decrease compared with the control sample are reported. Ethylene treated sample had higher content in ethyl hexanoate, above all at 24 and 36 h of treatment, with increases of around 190 and 216%, respectively (Table 3). Even ethyl decanoate and ethyl octanoate were in greater quantities after 36 h treatment compared to the control (+227 and +185, respectively), while in the other samples the values were more similar to the air sample (Table 3). Moreover, 36 h ethylene treatment stimulated a 10-fold increase in ethyl acetate synthesis over 15 and 24 h ethylene treatment; isoamyl acetate synthesis (+154%) was also stimulated in 36 h ethylene-treated grapes while the other ethylene treatments exhibited slight decreases over the air samples (Table 3). Together with ethyl acetate, acetic acid increased significantly in 36 h ethylene-treated grapes compared to the other samples (data not shown). Among alcohols, ethanol showed the same pattern as ethyl acetate, with an increase of 344% for 36 h ethylene treatment (Table 3). The rise of the other samples over the control sample was between 40 and 100%.

Comparing the data relative to ethanol, acetic acid, and ethyl acetate, it is possible to observe a concomitant increase in all of them, which would indicate an anaerobic process induced by the long-term ethylene treatment. The great content of acetic acid also induces the formation of isoamyl acetate, by esterification with the high isoamyl alcohol (3-methylbutan-1-ol) concentration, which exhibited the highest increase (186%) against the control kept in air. Manríquez et al. have shown the strong dependence of alcohol dehydrogenase (ADH) on ethylene in melons [21], and the ADH stimulation by ethylene in grapes was already observed [22]. It appears that 36 h ethylene treatment on Cesanese grapes accelerates the senescence process, triggering an anaerobic metabolism, as well as the catabolism of amino acids like leucine, which isoamyl alcohol comes from. Together with the rise in alcohol content, an increase in acetaldehyde (to a lesser extent), acetic acid and its ester, ethyl acetate, and even isoamyl acetate was observed. This means that ethanol, when it reaches the high amount observed, is oxidised back to acetic acid via acetaldehyde by means of ADH working in the ethanol-acetaldehyde direction, and acetaldehyde is very toxic for plant cells [23]. The formation of these metabolites might be the cause of polyphenols loss as the consequence of a strong oxidation process.

In air, C6 compounds such as hexanol, hexanal, and (E)-hex-2-enal did not change significantly, but in ethylene-treated samples the increase was significant for 24 and 36 h treatment, 27% and 20%, respectively, over the air sample (Table 2). In parallel with the acceleration of senescence and the shift to anaerobic metabolism, it is possible to assist to an oxidation process confirmed by the increase in C6 compounds, as previously observed in Aleatico [13] as well as by the ethanol oxidation. The C6 increase, above all with respect to the aldehydes, leads us to assume a stimulation of lipid peroxidation of linoleic and linolenic acids [24,25] via lipoxygenase activity [26]. Moreover, the increase in ethyl esters of fatty acids at intermediate carbon number with ethanol, such as ethyl hexanoate, ethyl octanoate, and ethyl decanoate, following ethylene treatment for 24 but above all 36 h, would confirm the degradation of the membrane lipid layer. In mango, ethylene stimulated the formation of fatty acids which significantly affected the fully ripe aroma [27]. Here, it is supposable to observe a similar turnover of fatty acids with higher synthesis and higher oxidation following ethylene treatment. In addition, we know that alcohol acyl transferase (AAT), key enzyme for esters formation, is ethylene-dependent, as described in apple [28] and in melon [29].

Regarding, terpene alcohols, long term treatment, 24 h but above all 36 h, significantly reduced their amount compared to air, with a percentage of decrease of 12% and 69%, respectively (Table 2). Geraniol decreased from 920,000 (peak area) in air-treated samples down to 210,000 in 36 h-treated sample, and citronellol from 250,000 down to 60,000 (data not shown). Among C13-norisoprenoids, the β-damascenone concentration was reduced by ethylene treatment compared to the control sample and to the initial value of 22%, 11%, and 44%, respectively for 15, 24, and 36 h ethylene treatments. (3). The strong oxidative environment in long term ethylene-treated samples would explain the loss of terpenols because these compounds are very sensitive to oxidation [30]. Even the decrease in β-damascenone can be attributed to this strong oxidative environment, since this compound synthesised, such as β-ionone, through the oxidation of carotenoids (breakdown), is itself rapidly oxidised [31,32]. All of the other detected volatile compounds (except for eugenol) present in the headspace of the containers where the grapes were maintained, such as octanol, nonanol, ethyl laurate, ethyl isovalerate, and ethyl lactate, progressively decreased in the ethylene-treated samples with the length of the treatment, going from 102,000 down to 69,000 (data not shown).

### Multivariate Observations

Original results coming from analytical measurements (in triplicate) for all quality parameters were also used to arrange a data matrix to perform a multivariate discrimination. Principal component analysis (PCA) was calculated considering each analytical variable as the loadings of the calculation, while the initial time, the ethylene (15, 24, and 36 h) treatments, and the untreated samples (AIR) represented the scores [33]. Just before to perform the chemometric procedure, data were autoscaled and row centred with the aim to make comparable data derived from different analysis and unit of measure. Five principal components (PCs) minimised the residual variance below the 5% (97.44% of explained variance), and calculated loadings (Table 4) evidenced significant effect of practically all measured VOCs on PC1 (38.2%), with decrescent influence of ethanol, isoamyl alcohol, ethyl hexanoate, acetic acid, β-damascenone, and isoamyl acetate, while C6 and terpenols, even positives, were less strong. All anthocyanins had a positive correlation on PC1, excluding malvidin-3-monoglucoside which demonstrated a negative correlation that confirmed on PC2 as well. Also sugars (°Brix) manifested opposite correlation on PC1, conversely to the titratable acidity and the pH. VOC influence was negative on PC2 (32.6%) where anthocyanins, indeed, presented great impact on the explained variance associated to the score segregation. Summarizing, the highest influence seems to be generated by the polyphenol metabolites which affected discrimination on both PC1 and PC2.

In graphical score plot (Figure 2A) and loading plot (Figure 2B), PC1 vsersus PC2 is plotted and there is well described, respectively, the bunching response of ethylene-treated and untreated grape samples affected by the response of the analytical variables. Initial time samples are well segregated from ethylene treated ones, moving along the two opposite 4th and 2nd Cartesian plan quadrants, by respecting the timing of treatments (Figure 2A). The 36 h treated samples are clearly well separated at the farther location, while 15 and 24 h are a little overlapped. The untreated samples (AIR) have moved to the 3rd quadrants and they are segregated from the other samples. Loadings (Figure 2B) evidenced how 36 h ethylene treated samples are mainly associated to the maximum influence of VOCs, while polyphenols and anthocyanins (total amount and specific compounds) are better linked to the 15 and 24 h treated. These responses, presented as pattern recognition mode, confirmed the evidences before discussed and underlined.

## 4. Conclusions

In conclusion, Cesanese grapes respond to ethylene treatment by increasing the polyphenol and anthocyanin content (up to a maximum level of about +23% and +17%, respectively) when the application is performed for a maximum time of exposure of 24 h. Beyond that time (36 h), a no evident effect on total polyphenols is manifested, while a detrimental consequence is observed in the anthocyanin decrease (−14%) and in the significant increase of ethanol, acetic acid, and ethyl acetate. Also, C6 compounds and fuel alcohols tend to increase under the action of the treatments. Ethylene treatment at high concentration can be used for short term application aimed at increasing the phenol content of Cesanese berry, confirming what has been already observed in Aleatico grapes in a paper previously published by our research group. As additional knowledge, in the current paper, the effects of different time exposures were better elucidated. In enological protocols, the treatments could be addressed to improve secondary metabolites extraction, just in case of wine grape varieties traditionally affected by problems of colour transfer. A quite significant novelty proposed by the paper can be found looking at the results of the performed multivariate observation (PCA). There, a significant separation among initial time, ethylene treated, and untreated samples was well appreciated, concomitantly describing the influence of all analytical parameters on that segregation. This would appear to be a good tool for not excluding any possible negative impact, in terms of an enological goal, while we are pushing for an improvement of polyphenol presence in the final wines, as in the case of ethylene treatments on wine grapes here proposed.

## Figures and Tables

**Figure 1 foods-10-00322-f001:**
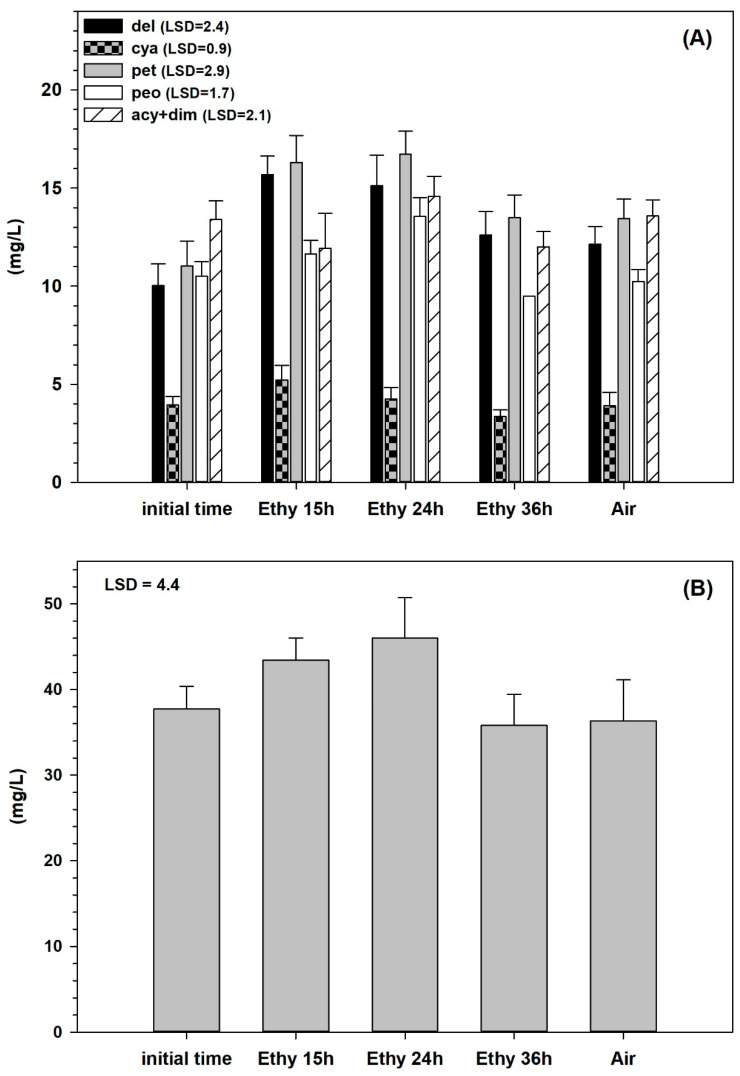
Anthocyanins content (mg L^–1^) in fresh sample and at the end of the experiment (36 h) in Cesanese grapes treated with ethylene for 15, 24, and 36 h or air for 36 h. Values represent means of three analyses from three different bunches ± SD. Calculated LSD (mg L^–1^) is per *p* = 0.05 (**A**). Malvidin content (mg L^−1^) in fresh sample and at the end of the experiment (36 h) in Cesanese grapes treated with ethylene for 15, 24, and 36 h or air for 36 h. Values represent means of three analyses from three different bunches ± SD. Calculated LSD (mg L^–1^) is per *p* = 0.05 (**B**).

**Figure 2 foods-10-00322-f002:**
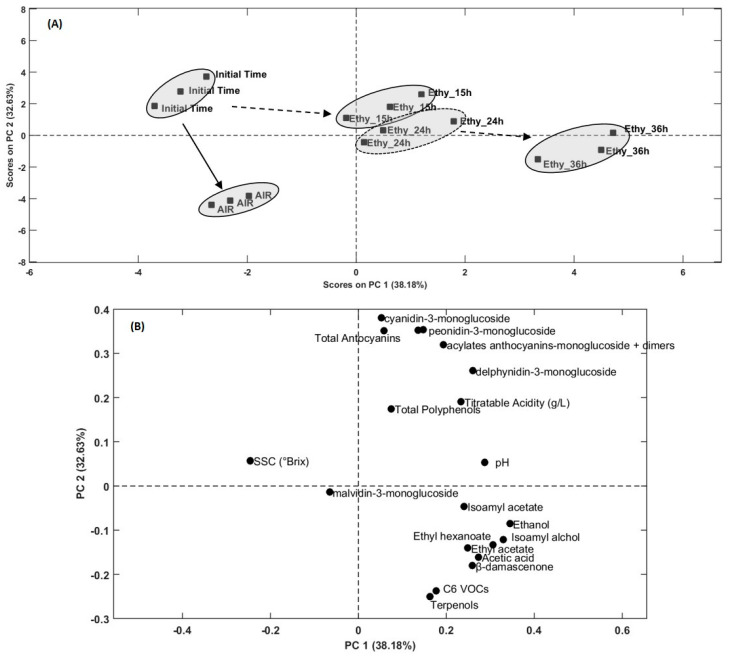
Principal component analysis (PCA) score plot (**A**, top) and loading plot (**B**, bottom) including graphical representations of PC1 vs. PC2. Scores included in the plot (**A**) are not scaled on loading values.

**Table 1 foods-10-00322-t001:** Schematic protocol of treatments. Time of treatments represents the hours of grape permanence at treatment conditions (ethylene or air) inside the chambers. Time after treatments is the time of prolonged grape permanence inside the chambers, over treatment conditions, for reaching the same total time of 36 h.

	Air-Treated	Ethylene-Treated (500 mg L^−1^)		
**Time of treatment (h)**	36	15	24	36
**Time after treatment (h)**	0	21	12	0
**Total time (h)**	36	36	36	36

**Table 2 foods-10-00322-t002:** Initial and final (36 h) values of soluble solids content (SSC) (°Brix), pH, titratable acidity (TA), total polyphenols, total anthocyanins, and anthocyanins/polyphenols (A/P) ratio in ethylene- (15, 24, and 36 h) treated samples over air-treated one. Data are the means of the berries picked from three different bunches each sampling (30 berries for SSC, 3 reps for the remaining analyses). Mean separation was performed by applying least significant difference (LSD) test.

	SSC	pH	TA	Polyphenols	Anthocyanins	A/P Ratio
	(° Brix)		(g L^−1^)	(mg L^−1^ of Catechins)	(mg L^−1^)	(%)
**Initial time**	23.2 ± 0.7	3.7 ± 0.04	4.5 ± 0.08	412 ± 26	160 ± 9	39
**Ethylene 15 h**	21.5 ± 0.5	4 ± 0.01	4.4 ± 0.03	505 ± 33	187 ± 12	37
**Ethylene 24 h**	21.6 ± 0.7	4 ± 0.04	4.6 ± 0.04	488 ± 22	150 ± 10	31
**Ethylene 36 h**	21.4 ± 0.5	4.1 ± 0.02	4.7 ± 0.03	405 ± 31	138 ± 8	34
**Air (control)**	22.2 ± 0.7	3.8 ± 0.04	4.2 ± 0.06	407 ± 28	107 ± 9	26
**LSD (*p* = 0.05)**	0.9	0.3	0.4	32	16	-

**Table 3 foods-10-00322-t003:** Percentage of increase or decrease (–) of main selected volatile organic compounds (VOCs) in ethylene-(15, 24, and 36 h)-treated samples over air-treated sample at the end of 36 h treatment. C6 compounds include hexanol, hexanal, and (E)-hex-2-enal. Reported values are calculated on original data which, in turn, were derived from the mean of three gas chromatography (GC) runs, each one made with grape juice obtained from three different bunches.

	Ethylene 15 h	Ethylene 24 h	Ethylene 36 h
**Ethanol**	40	107	344
**Ethyl acetate**	108	159	1192
**Isoamyl acetate**	−7	−17	154
**Isomyl alcohol**	−6	0	186
**Ethyl exanoate**	19	190	216
**Ethyl octanoate**	8	13	147
**Ethyl decanoate**	11	26	105
**C6 volatiles**	7	27	20
**Terpenols**	3	−12	−69
**β-damascenone**	−22	−11	−44

**Table 4 foods-10-00322-t004:** PCA loadings values for the five principal components (PCs) describing residual explained variance at the 95% of the accuracy.

	PC1 (38.17%)	PC2 (32.63%)	PC3 (12.44%)	PC4 (8.58%)	PC5 (5.59%)
**Ethanol**	0.95257	−0.21775	−0.18708	0.051498	0.033167
**Acetic acid**	0.75478	−0.41184	−0.24462	0.41223	0.14963
**Ethyl acetate**	0.68753	−0.35888	−0.20985	0.26748	−0.25038
**Isoamyl acetate**	0.66541	−0.11647	−0.13765	−0.61637	0.032228
**Isoamyl alchol**	0.91109	−0.30992	−0.1482	0.20466	0.059631
**Ethyl hexanoate**	0.84698	−0.33996	−0.095672	−0.38025	0.013219
**C6 VOCs**	0.48993	−0.60472	0.44466	−0.43387	−0.013936
**Terpenols**	0.45121	−0.63937	0.44359	0.42655	0.01025
**β−damascenone**	0.71723	−0.4603	0.4576	0.23179	−0.068807
**Delphynidin-3-monoglucoside**	0.71817	0.66607	−0.19051	0.046359	−0.02245
**Cyanidin-3-monoglucoside**	0.144	0.97288	0.089111	0.13606	−0.069622
**Petunidin-3-monoglucoside**	0.40832	0.90318	−0.0096921	0.040272	−0.12374
**Peonidin-3-monoglucoside**	0.37527	0.90096	−0.021239	0.20378	−0.062239
**Acylates anthocyanins-monoglucoside + dimers**	0.53409	0.81737	−0.072474	0.19128	−0.05762
**Malvidin-3-monoglucoside**	−0.17914	−0.034189	0.88987	0.35242	0.1997
**SSC (°Brix)**	−0.67895	0.14707	−0.069865	0.066328	0.69161
**pH**	0.79344	0.1364	0.36137	−0.10514	0.44036
**Titratable Acidity**	0.64479	0.48599	−0.22467	−0.16816	0.50142
**Total Polyphenols**	0.20811	0.44678	0.71218	−0.46353	−0.16587
**Total Antocyanins**	0.16156	0.89742	0.40293	0.02484	−0.060997

## Data Availability

The data presented in this study are available on request from the corresponding author.

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
