# Peer review of "Time of Postharvest Ethylene Treatments Affects Phenols, Anthocyanins, and Volatile Compounds of Cesanese Red Wine Grape"

_foods, 2021, doi:10.3390/foods10020322_

Round 1

Reviewer 1 Report

Line 37. 1-MCP, first full name, than abbreviation further in the text.

Line 59-61. Few words about Cesanese yield, SSC and TA?

Line 60. low polyphenols content

Line 66. vineyard instead of orchard. Could you provide few more information about vineyard?

Line 67. what do you mean "sound clusters"?

Line 68. harvested or picked instead of cut.

2.1. Test Material - could you provide any information about yield parameters of experimenatl variety, such as cluster or berry weight?

Line 89. RI, first full name. Randomly chosen bunches? It would be better if you use only clusters or bunches, not both.

Lines 90-92. could you please describe these methods in a few more details since they are not standard O.I.V. methods. Or just cite the paper that explain the method in details.

175. polyphenols and anthocyanin synthesis

181. why not shown data? And where are initial aromatic compunds?

182. it would be better synthesis than production

222. described

236. start a new paragraph with the "Original". Discuss PCA analysis separately, it would be more clear.

244. VOCs, volatile compounds? Full name when first appearance in the text.

276. State and explain more clearly why did you perform this experiment.  You already performed ethylen effects on Aleatico grapes in 2006 and in 2011. Is the only novelity 24h and 36h of treatment? If so, why is it so important? Why did you decide to examine those effects?

In general, the paper is well written. The level of English is good. The methods and statistical analysis appear to be adequate. The manuscript represents a nice addition to our understanding of postharvest effects of ethylene treatments. However, the paper is very similar to some previous papers of the same research group, so it is necessary to better emphasize the significance of this experiment. 

Author Response

Line 37. 1-MCP, first full name, than abbreviation further in the text.

AMENDED

Line 59-61. Few words about Cesanese yield, SSC and TA?

SINCERELY, WE DON’T UNDERSTAND THIS COMMENT. AT THAT POINT OF THE MANUSCRIPT (INTRODUCTION SECTION), WE DON’T SEE ANY USEFULL CORRELATION BETWEEN THE DESCRIPTION OF GENERAL CHARACTERS OF CESANESE GRAPE AND THE DEFINITION OF ITS DETAILED QUALITY ATTRIBUTES THAT WE FOLLOW REPORTED IN M&M SECTION   

Line 60. low polyphenols content

AMENDED

Line 66. vineyard instead of orchard. Could you provide few more information about vineyard?

ORCHARD HAS BEEN REPLACED BY VINEYARD. A MORE DETAILED DESCRIPTION OF THE VINEYARD AND OF THE VINE CULTIVATION HAS BEEN ADDED.

Line 67. what do you mean "sound clusters"?

WE CHANGED CLUSTERS IN BUCHES TRHOUGOUT ALL THE TEXT. SOUND MEANS WHOLESOME, INTACT AND WITH NO PATHOGEN ATTACTS. SOUND IS A QUALIFIER COMMONLY USED FOR FRUITS AND VEGETABLES IN SCIENTIFIC CONTEXT

Line 68. harvested or picked instead of cut.

AMENDED

2.1. Test Material - could you provide any information about yield parameters of experimenatl variety, such as cluster or berry weight?

UNFORTUNATELY, WE DIDN’T EVALUATE BUNCH AND/OR BERRY WEIGHT

Line 89. RI, first full name. Randomly chosen bunches? It would be better if you use only clusters or bunches, not both.

REFRACTOMETRIC INDEX (RI) WAS ALREADY REPORTED BEFORE (LINE 67). AMENDED, CLUSTERS HAS BEEN REMOVED AND ONLY BUCHES HAS BEEN USED.

Lines 90-92. could you please describe these methods in a few more details since they are not standard O.I.V. methods. Or just cite the paper that explain the method in details.

ADDED

  1. polyphenols and anthocyanin synthesis

AMENDED

  1. why not shown data? And where are initial aromatic compunds?

AMONG ESTER CATEGORY OF VOCS WE DECIDED TO SELECT THE MOST SIGNIFICANT, AS WELL AS THE OTHER CATEGORIES WERE GROUPED (e.g. C6 COMPOUNDS, AND TERPENOLS) AND SO REPORTED. CONSIDERING WE ARE NOT TREATING OF WINE SAMPLES, THE PRESENCE OF ESTERS IN GRAPE MUST IS DUE TO THE BONDING ACTION BETWEEN ALCOHOL AND ACIDS AND IT IS A CONSEQUENCE OF THE POSTHARVEST TREATMENT ON GRAPES AS WE CLARIFIED AND DISCUSSED INTO THE PAPER.

  1. it would be better synthesis than production

AMENDED

  1. described

CORRECTED

  1. start a new paragraph with the "Original". Discuss PCA analysis separately, it would be more clear.

DONE

  1. VOCs, volatile compounds? Full name when first appearance in the text.

AMENDED

  1. State and explain more clearly why did you perform this experiment.  You already performed ethylen effects on Aleatico grapes in 2006 and in 2011. Is the only novelity 24h and 36h of treatment? If so, why is it so important? Why did you decide to examine those effects?

THANKS FOR YOUR SUGGESTIONS. SEVERAL CHANGES HAVE BEEN MADE IN THE INTRODUCTION  AND, PARTICULARLY, THE CONCLUSION SECTION HAS BEEN SIGNIFICANTLY IMPROVED JUST FOR ELUCIDATING ALL PENDING DOUBTS AND OBSERVATIONS

Reviewer 2 Report

  • Please, define 1-MCP and SSC the first time they are mentioned in the text

  • My major concern is about the methods of analysis used:
    • HPLC-PDA analysis to determine the anthocyanins content: the information of the chromatographic column is shown but authors should provide information about the main chromatographic parameters (eluents, isocratic/gradient elution, flow, time of analysis, etc) and about the detector conditions (selected wavelenght/s). In line 98 authors indicate that this analysis provided qualitative information, however in the discussion they use the data to evaluate tendencies (increase or decrease of the content of the analytes). Therefore, I suggest using the term semiquantitative instead of qualitative. Also, it would be very interesting to include a comment about the reproducibility of the method.
    • GC analysis to obtain the aroma profile: this method was already developed by authors and slightly modified. It should be interesant to include in the manuscript some information about the quality parameters of the method such as linearity, trueness, precision and limits of quantification.
    • In line 120, the use of standards is mentioned but the name of these standards is not included in the text. 

  • Tables 1 and 2. As values of the different parameters evaluated are obtained after the analysis of three independent replicates, the corresponding errors should be provided. 

  • Figure 1. The error bars should be included in the two graphics

Author Response

  • Please, define 1-MCP and SSC the first time they are mentioned in the text

 DONE

  • My major concern is about the methods of analysis used:
    • HPLC-PDA analysis to determine the anthocyanins content: the information of the chromatographic column is shown but authors should provide information about the main chromatographic parameters (eluents, isocratic/gradient elution, flow, time of analysis, etc) and about the detector conditions (selected wavelenght/s). 
    • THANK YOU FOR THE COMMENT CONCERNING THE DETAIL ON THE HPLC METHOD AND PROTOCOL. IT HAS BEEN USEFUL FOR REALIZING THAT THERE WERE SEVERAL LACKES AND MISTAKES. WE APPORTED ALL THE ADEQUATE MODIFICATIONS, AS WELL AS WE BETTER ELUCIDATED THE DESCRIPRION OF ADOPTED HPLC PROTOCOL
    • In line 98 authors indicate that this analysis provided qualitative information, however in the discussion they use the data to evaluate tendencies (increase or decrease of the content of the analytes). Therefore, I suggest using the term semiquantitative instead of qualitative. Also, it would be very interesting to include a comment about the reproducibility of the method.
    • AMENDED
    • GC analysis to obtain the aroma profile: this method was already developed by authors and slightly modified. It should be interesant to include in the manuscript some information about the quality parameters of the method such as linearity, trueness, precision and limits of quantification.
  • AS YOU WELL OBSERVED, THE GC/SPME METHOD WAS ALREADY DEVELOPED, POINTED OUT AND VALIDATED BY OUR RESEARCH GROUP IN SIMILAR PUBLISHED PAPERS WHERE THE ANALYTICAL PROTOCOL WAS QUITE WELL DESCRIBED. THE GOAL OF ALL THESE SCIENTIFIC MANUSCRIPTS IS VERY SIMILAR: A FINAL COMPARATION AMONG SAMPLES ADDRESSED TO DEFINE POSSIBLE DIFFERENCES IN VOLATILES DUE TO THE EFFECTS OF THE POSTHARVEST TREATMENTS ON WINE GRAPES, AND THIS WAS ACHIEVED BY THE PROPOSED INSTRUMENTAL APPROACH. ON THIS BASIS, WE CONSIDERED THAT THE DETAILS FOUNDABLE IN CITED ARTICLE CAN BE CONSIDERED AS A GOOD METHODOLOGICAL TOOL FOR READERS
    • In line 120, the use of standards is mentioned but the name of these standards is not included in the text. 
    • THE HPLC STANDARDS ARE ALREADY MENTIONED; A SHORT SENTENCE HAS BEEN ADDED WITH RESPECT TO THE GC STANDARDS

 Tables 1 and 2. As values of the different parameters evaluated are obtained after the analysis of three independent replicates, the corresponding errors should be provided. 

AMENDED

 Figure 1. The error bars should be included in the two graphics

AMENDED

Reviewer 3 Report

It is a very interesting manuscript reporting on the postharvest treatment of wine grapes with ethylene to improve some key parameters desired by wine-makers.  It follows (to a large extend) the earlier work (ref. [13]) with some additional insights and hypotheses drawn upon the application of the same approach to a different grape variety.  Overall, the manuscript has a lot of merits, but it needs more attention to organization, details, and readability.   Specific comments for improvement are below:

Abstract:

L15 – please add the relative percent increase and statistical significance (e.g., p-value)

L16-20 - please add the relative percent increase and statistical significance (e.g., p-value).

Keywords: please try not to repeat the words from the title and making them broader (e.g., postharvest treatment, fruit ripening, phytohormone, etc.)

Introduction:

L51-55 – please revise this long sentence.

Materials and Methods:

L74-79 – please convert this into a Table. It will be much more accessible to readers.  Please label treatments in some easy-to-follow fashion here and in the Results.

L84 – please provide more specific information about the grade (purity of ethylene) in the gas cylinder.

It would be benefit readers if the Authors provided a figure schematic on the gas delivery system, cylinder with grapes, key parts, gas and airflow, since the information in [13] is all text.

L103 – lowercase ‘gas chromatography.’

L114 – add space between numbers and units (here and elsewhere in the manuscript – many instances of the same formating issue).

Results:

Figure 1 – please add error bars.

L176 – new paragraph here.

Table 2 – please add statistical analyses here or in the text.

L209, 222, 236  – new paragraph here.

Conclusions – please the statistical analysis results there, similarly to the improvements made in the Abstract.

Author Response

L15 – please add the relative percent increase and statistical significance (e.g., p-value)

WE ADDED THE PERCENTAGE INCREASE; WE BELIEVE IT IS ENOUGH FOR THE ABSTRACT. STATISTICAL SIGNIFICANCE CAN BE APPRECIATED IN DETAIL INTO THE TEXT

L16-20 - please add the relative percent increase and statistical significance (e.g., p-value).

DONE. LOOK AT THE ANSWER BEFORE

Keywords: please try not to repeat the words from the title and making them broader (e.g., postharvest treatment, fruit ripening, phytohormone, etc.)

SORRY FOR DISAGREING WITH YOUR CONSIDERATION, BUT IN OUR OPINION KEYWORDS ARE WELL REPRESENTATIVE OF THE MS CONTENT

Introduction:

L51-55 – please revise this long sentence.

DONE

Materials and Methods:

L74-79 – please convert this into a Table. It will be much more accessible to readers.  Please label treatments in some easy-to-follow fashion here and in the Results.

HONESTLY, WE BELIEVE THE SCHEMATIC REPRESENTATION IS SUFFICIENTLY CLEAR AND REPRESENTATIVE OF THE PROTOCOL FOR BEING WELL APPRECIATED BY THE READERS. ADDITIONALLY, THE TWO OTHER REVIEWERS DIDN’T REQUEST THIS ADDITIONAL TABLE

L84 – please provide more specific information about the grade (purity of ethylene) in the gas cylinder.

WE ADDED SOME DETAILS CONCERNING THE ETHYLENE PURITY AND THE ADOPTED PROTOCOL FOR GASEOUS TREATMENTS (LOOK AT THE LINES 86-87 OF THE REVISED VERSION). IN OUR OPIONION NO ADDITIONAL INFORMATIONS AND/OR A SCHEMATIC FIGURE OF THE GAS DELIVERY SISTEM ARE NEEDED. IN EQUIVALENT ARTICLES, NO SIMILAR PRESENTATION HAS BEEN FOUND, WHEN THE SCIENTIFIC AIM IS ADDRESSED TO DESCRIBE THE PHYSIOLOGICAL AND METABOLIC EFFECT OF THE ETHYLENE TREATMENT….

It would be benefit readers if the Authors provided a figure schematic on the gas delivery system, cylinder with grapes, key parts, gas and airflow, since the information in [13] is all text.

PLEASE, LOOK AT THE UPPER ANSWER

L103 – lowercase ‘gas chromatography.’

AMENDED

L114 – add space between numbers and units (here and elsewhere in the manuscript – many instances of the same formating issue).

AMENDED

Results:

Figure 1 – please add error bars.

ADDED

L176 – new paragraph here.

AMENDED

Table 2 – please add statistical analyses here or in the text.

STATISTICAL ANALYSIS? IT IS ABOUT OF CALCULATED PERCENTAGE OF INCREASE (+) OR DECREASE (-) BASED ON THE MEAN OF THE ORIGINAL VALUES, AS IT IS REPORTED IN THE TABLE CAPTION

L209, 222, 236 – new paragraph here.

AMENDED

Conclusions – please the statistical analysis results there, similarly to the improvements made in the Abstract.

THE CONCLUSION SECTION HAS BEEN STRONGLY REVISED AND MODIFIED, HOPING TO HAVE SIGNIFICANTLY IMPROVED ALL PENDING DOUBTS AND OBSERVATIONS

Round 2

Reviewer 2 Report

Thank you very much for amending the manuscript and considering my suggestions

Reviewer 3 Report

The manuscript has been trending in the right direction.  The Authors made improvements which can be appreciated.  However, there is this unnecessary push-back on easy-to-do items that, in the end, would benefit both the readers and Authors. 

Specifically, from the R1:

(1) Reviewer - Keywords: please try not to repeat the words from the title and making them broader (e.g., postharvest treatment, fruit ripening, phytohormone, etc.)

Authors - SORRY FOR DISAGREING WITH YOUR CONSIDERATION, BUT IN OUR OPINION KEYWORDS ARE WELL REPRESENTATIVE OF THE MS CONTENT

Additional comment for R2 - this is just a good practice for manuscript writing.

(2) Reviewer - L74-79 – please convert this into a Table. It will be much more accessible to readers.  Please label treatments in some easy-to-follow fashion here and in the Results.

Authors - HONESTLY, WE BELIEVE THE SCHEMATIC REPRESENTATION IS SUFFICIENTLY CLEAR AND REPRESENTATIVE OF THE PROTOCOL FOR BEING WELL APPRECIATED BY THE READERS. ADDITIONALLY, THE TWO OTHER REVIEWERS DIDN’T REQUEST THIS ADDITIONAL TABLE.

Additional comment for R2- this is not a good practice to deflect to other reviewers.  We (reviewers) are trying to help from different types of experience. We do not have to agree.  In the end, the quality of the manuscript is what matters.

(3) Reviewer - It would be benefit readers if the Authors provided a figure schematic on the gas delivery system, cylinder with grapes, key parts, gas and airflow, since the information in [13] is all text.

Authors - PLEASE, LOOK AT THE UPPER ANSWER - cited here as follows - 'WE ADDED SOME DETAILS CONCERNING THE ETHYLENE PURITY AND THE ADOPTED PROTOCOL FOR GASEOUS TREATMENTS (LOOK AT THE LINES 86-87 OF THE REVISED VERSION). IN OUR OPIONION NO ADDITIONAL INFORMATIONS AND/OR A SCHEMATIC FIGURE OF THE GAS DELIVERY SISTEM ARE NEEDED. '

Additional comment for R2 - there is plenty of opportunity to present this information to the benefit of readers - especially, since the novelty and significance are claimed.  Supplemental Materials format could be used to show the schematics if there is not room in the main manuscript.

(4) Reviewer - Table 2 – please add statistical analyses here or in the text.

STATISTICAL ANALYSIS? IT IS ABOUT OF CALCULATED PERCENTAGE OF INCREASE (+) OR DECREASE (-) BASED ON THE MEAN OF THE ORIGINAL VALUES, AS IT IS REPORTED IN THE TABLE CAPTION.

Additional comment for R2 -the statistical analyses need to be reported for the benefit of readers.  The table caption can be edited.

Author Response

(1) Reviewer - Keywords: please try not to repeat the words from the title and making them broader (e.g., postharvest treatment, fruit ripening, phytohormone, etc.)

Authors - SORRY FOR DISAGREING WITH YOUR CONSIDERATION, BUT IN OUR OPINION KEYWORDS ARE WELL REPRESENTATIVE OF THE MS CONTENT

Additional comment for R2 - this is just a good practice for manuscript writing.

THANK YOU VERY MUCH, WE ARE SURE THAT YOUR INDICATION IS ADDRESSED TO IMPROVE THE GENERAL READABILITY AND QUALITY OF THE MANUSCRIPT. HOWEVER, THE REPORTED KEYWORDS FIT WELL, IN OUR OPINION, WITH THE PAPER, AND THIS SHOULD BE THE AIM OF THE KEYWORDS. THE FACT THAT THERE IS AN OVERLAPPING WITH THE WORDS INCLUDED IN THE TITLE SHOULD BE CONSIDERED A GOOD ELEMENT FOR THE STRONGNESS AND REPRESENTATIVITY OF THE SAME TITLE.

(2) Reviewer - L74-79 – please convert this into a Table. It will be much more accessible to readers.  Please label treatments in some easy-to-follow fashion here and in the Results.

Authors - HONESTLY, WE BELIEVE THE SCHEMATIC REPRESENTATION IS SUFFICIENTLY CLEAR AND REPRESENTATIVE OF THE PROTOCOL FOR BEING WELL APPRECIATED BY THE READERS. ADDITIONALLY, THE TWO OTHER REVIEWERS DIDN’T REQUEST THIS ADDITIONAL TABLE.

Additional comment for R2- this is not a good practice to deflect to other reviewers.  We (reviewers) are trying to help from different types of experience. We do not have to agree.  In the end, the quality of the manuscript is what matters.

FOLLOWING YOUR SUGGESTION, WE CREATED A NEW TABLE (ACTUALLY TABLE 1) WERE THE SCHEMATIC REPRESENTATION OF THE TREATMENT PROTOCOL HAS BEEN REPORTED. SINCE SOME DETAILS OF THE OPERATIONS RELATED TO THE TREAMENTS CAN BE WELL EXPLAINED ONLY THROUGH AN EXTENDED TEXT, WE ARE NOW IN DOUBT IF THE WRITTEN PART MUST TO BE TOTALLY REMOVED. AT THE MOMENT, IT IS STILL PRESENT, AND THE REQUESTED TABLE HAS BEEN ADDED AS AN ADDITIONAL INFORMATION.   

(3) Reviewer - It would be benefit readers if the Authors provided a figure schematic on the gas delivery system, cylinder with grapes, key parts, gas and airflow, since the information in [13] is all text.

Authors - PLEASE, LOOK AT THE UPPER ANSWER - cited here as follows - 'WE ADDED SOME DETAILS CONCERNING THE ETHYLENE PURITY AND THE ADOPTED PROTOCOL FOR GASEOUS TREATMENTS (LOOK AT THE LINES 86-87 OF THE REVISED VERSION). IN OUR OPIONION NO ADDITIONAL INFORMATIONS AND/OR A SCHEMATIC FIGURE OF THE GAS DELIVERY SISTEM ARE NEEDED. '

Additional comment for R2 - there is plenty of opportunity to present this information to the benefit of readers - especially, since the novelty and significance are claimed.  Supplemental Materials format could be used to show the schematics if there is not room in the main manuscript.

FOLLOWING YOUR INDICATION, WE HAVE DRAWN A GRAPHICAL REPRESENTATION OF THE ADOPTED TREATMENTS. WE SUGGEST THAT THE SCHEME CAN BE INCLUDED AS SUPPLEMENTAL MATERIAL. THANKS   

(4) Reviewer - Table 2 – please add statistical analyses here or in the text.

STATISTICAL ANALYSIS? IT IS ABOUT OF CALCULATED PERCENTAGE OF INCREASE (+) OR DECREASE (-) BASED ON THE MEAN OF THE ORIGINAL VALUES, AS IT IS REPORTED IN THE TABLE CAPTION.

Additional comment for R2 -the statistical analyses need to be reported for the benefit of readers.  The table caption can be edited.

NO STATISTICAL INDEXES CAN BE ADD TO DATA CALCULATED AS A PERCENTAGE OF INCREASE OR DECREASE, ON THE AVERAGED VALUES OF GC DETECTIONS. WE HOPE TO HAVE INCLUDED IN THE TABLE CAPTION ALL THE NEEDED DETAILS. MANY THANKS
